**www.cambridge.org/gmh**

scalable; refugees; psychological; transdiagnostic

**Author for correspondence:**
Richard A. Bryant,
Email: r.bryant@unsw.edu.au

# Scalable interventions for refugees

## Richard A. Bryant 

School of Psychology, University of New South Wales, Sydney, NSW, Australia

### Abstract

Refugees experience a greater rate of common mental disorders relative to most other populations, and there remains a need to address these needs. However, most refugees are hosted in low-and-middle-income countries, where there is a lack of resources and mental health providers who can deliver mainstream mental health services. This situation has led to the emergence of scalable mental health interventions that can deliver evidence-based programs to refugees in need. Many countries hosting refugees have implemented programs that train local lay providers in interventions that can be delivered at scale. This review provides a narrative overview of these scalable interventions and critiques the evidence for their efficacy. It is noted that there are limitations to currently available scalable interventions, and there is a need for greater attention to determining the longer-term benefits of interventions, addressing the mental health needs of refugees who do not respond to these interventions, assisting refugees with more severe psychological disorders, and understanding the specific mechanisms that underpin observed benefits of these interventions.

### Impact statement

There is a significant gap between mental health needs and availability of mental health services in many countries, especially low-and-middle-income countries. To address this gap, there has been the emergence of interventions that can be scaled up in settings that lack mental health services. These scalable interventions often utilise lay providers who receive brief training, and can offer programs that are typically based on cognitive behaviour principles. The available evidence points to these interventions being moderately effective in reducing common mental health problems, such as anxiety and depression. These interventions offer governments and agencies the opportunity to use individual, group and digital programs to address the mental health needs of people in need. Despite the potential of these programs to assist people, there should also be caution in implementing these programs because cost-effectiveness, real-world implementation and understanding the active ingredients of these interventions have yet to be evaluated.

### Introduction

Refugees are at risk of higher rates of mental disorders than many other populations because of the nature of the prolonged traumatic experiences they can endure. There is considerable evidence that many types of psychological interventions are effective in mitigating the symptoms of common mental disorders, including anxiety, depression and posttraumatic stress disorder (PTSD) (Cuijpers et al., 2021; McLean et al., 2022). However, the vast majority of this evidence comes from studies conducted in high-income countries that have the luxury of well-resourced health infrastructures and mental health specialists. This can be problematic for addressing the mental health needs of most of the world's refugees because most refugees are hosted in low-and-middle-income countries (LMICs), where there are typically inadequate resources to provide specialist mental health services. This situation has led to a shift to develop interventions that can both reduce psychological disorders in LMICs, and also be sustainable and scalable in these settings. This review provides an overview of attempts to develop psychological interventions that can be scaled up in LMICs to mitigate the mental health problems of refugees. Rather than providing a systematic review of scalable interventions, which has been reported previously (Singla et al., 2017), this review commences with a description of the major types of scalable interventions currently being offered to refugees, and offers a critique of the available evidence regarding these interventions, with the view of understanding how these interventions can benefit refugees, and also points to future challenges for this relatively new field.

### The need for scalable interventions

The need for effective mental health interventions for refugees is indicated by the strong evidence that refugees have higher rates of mental disorders than many community samples. One meta-

analysis of refugees re-settled in high-income countries reported PTSD prevalence of 29% when assessed via clinical diagnosis and 37% based on self-report measures (Henkelmann et al., 2020). Other meta-analyses have noted rates of PTSD in approximately 30% of refugees (Steel et al., 2009; Blackmore et al., 2020). These rates are higher than observed rates in community samples reported in the World Mental Health Survey which found a cross-national rate of 3.9% (Koenen et al., 2017). A number of studies have focused on specific refugee groups, with meta-analyses of Syrian refugees indicating between 31 and 43% of Syrians re-settled in other countries reporting PTSD (Peconga and Hogh Thogersen, 2019; Nguyen et al., 2022). Refugees also commonly experience depression, anxiety and suicidality. One meta-analysis found prevalence rates of depression and anxiety were 31.5 and 11%, respectively (Blackmore et al., 2020). Another meta-analysis of refugee studies reported elevated pooled prevalence rates for PTSD (31%), major depression (32%), and importantly showed that rates of depression were higher in LMIC (Patanè et al., 2022). An umbrella review considered five systematic reviews and concluded that rates of depression and anxiety were somewhat higher than rates of PTSD, with point estimates being 4–40% for anxiety, 5–44% for depression and 9–36% for PTSD (Turrini et al., 2017). Refugees also experience a number of other potentially severe psychological disorders. Prolonged Grief Disorder can be common in refugees as a result of the many traumatic bereavements they can experience (Tay et al., 2016). Representative studies of refugees reported estimated prevalence of 15% of bereaved refugees from various backgrounds in a high-income country (Bryant et al., 2019), and a similar rate has been reported in Syrian refugees in a refugee camp in Jordan (Bryant et al., 2021).

One of the major problems facing most of the world's refugees is that they are hosted in LMICs. This is common because these countries are adjacent to the homeland from which refugees have fled, and are therefore more easily accessed. For example, the vast majority of Syrian refugees are hosted in Türkiye, Lebanon, Jordan and Iraq. This trend can exacerbate many of the mental health needs of refugees because these settings typically have few mental health specialists and limited budgets for mental health care This situation can result in a treatment gap between mental health needs and provision of services. Depending on a country's income level, between country income, between 7% and 28% of people with depression receive treatment; overall, about one-third of cases in LMICs receive treatment compared to more than half of cases in high-income countries (Chisholm et al., 2016). One review found that whereas 36.3% of respondents in high-income countries in the World Mental Health Survey who reported an anxiety disorder received help, only 13% of those in LMICs who reported an anxiety disorder reported receiving assistance (Alonso et al., 2018). Another meta-analysis of global studies found a significant gap in mental health service use, which ranged from 33% of those needing help actually receiving it in high-income countries to 8% in LMICs (Moitra et al., 2022). This meta-analysis also reported that whereas 23% of people received minimally adequate treatment, only 3% received this level of care in LMICs. It is for these reasons that the Lancet Commission on Global Mental Health and Sustainable Development Goals emphasised the significant gap between mental health needs in poorly resourced countries and the availability of services as one of the key public health issues confronting LMICs (Patel et al., 2018).

The lack of mental health services is not the only barrier to people accessing mental health care in LMICs. Many refugees have low rates of treatment-seeking, which can be attributed to limited financial resources, low availability of interpreters or difficulties with transport or managing competing responsibilities such as childcare (Slewa-Younan et al., 2014). Many refugees also have negative beliefs about receiving help for mental health problems, which limits their motivation for seeking help that may be available (Byrow et al., 2020). Different refugee groups hold distinct conceptualizations of mental health, and often the hosting country's mental health system does not accord with the normative expressions of mental health that are held in the refugee's own culture. Further, some refugees have had personal experiences that lead to mistrust of government or agency services, and this can contribute to reluctance to seek mental health care (Nickerson et al., 2014). It is also very important to note that many refugees have their own means of coping with psychological difficulties, which can often involve adaptive use of social, communal, and religious methods. The reliance on these strategies may serve an important function in buffering the effects of psychological distress, and therefore minimize reliance on mainstream health services.

## Do evidence-based treatments work in LMICs?

There is much evidence that psychological treatments developed in high-income countries can work effectively in LMIC. For example, in the case of PTSD, trauma-focused psychotherapies have been studied often in high-income contexts, typically involving techniques that encourage emotional processing of trauma memories and restructuring cognitive appraisals about the trauma (Forbes et al., 2020). Meta-analyses indicate that there is moderate quality evidence this approach can reduce PTSD, as well as anxiety and depression, in refugees in both high-income and LMICs (Turrini et al., 2019), which supports the conclusion of other meta-analyses that show that this approach is effective in LMICs to alleviate PTSD in refugees in high-income countries (Nose et al., 2017) and refugees generally (Thompson et al., 2018). Importantly, meta-analyses that focus exclusively on studies of people in LMICs also show that evidence-based psychotherapy for PTSD can effectively reduce PTSD in these countries (Morina et al., 2017). Some variants of evidence-based programs have been specifically developed for LMICs, and especially for refugees. One popular variant of trauma-focused psychotherapy is narrative exposure therapy (NET), which is adapted from prolonged exposure therapy to address the needs of refugees but also builds on testimony psychotherapy to direct the person to emotionally process trauma memories and also integrate other positive aspects of their life to provide a narrative of their life's story (Schauer et al., 2005). This approach allows the multiple traumatic events that many refugees experience, as well as the considerable other forms of adversity, to be processed and interspersed with positive memories that are part of their autobiographical histories. There are multiple trials of NET working effectively with refugees in Uganda (Neuner et al., 2004, 2008; Ertl et al., 2011) in LMICs. Meta-analyses have also concluded that this approach is efficacious for treating PTSD (Lambert and Alhassoon, 2015; Nose et al., 2017). The extent to which NET is to be considered a scalable intervention has yet to be fully established, however. Although its efficacy is indicated, the NET protocol suggests that it should be delivered across four to 12 sessions (Schauer et al., 2011), which may represent a longer intervention than is feasible in many low-resource settings. This issue is underscored by meta-analytic evidence that the efficacy of NET increases with the number of sessions offered (Lambert and Alhassoon, 2015).

## The emergence of scalable interventions

Despite the efficacy of psychological interventions in high-income countries, there is little evidence that these have been scaled up to the point of broad implementation in health systems in LMICs. This has not occurred because these evidence-based interventions typically require mental health specialists, most treatment programs are disorder-specific which results in health providers needing to master multiple programs and be trained in complex differential diagnosis procedures, and the majority of evidence-based treatments for common mental disorders require at least 10 sessions, which is costly for health services and demanding on recipients (Eaton et al., 2011). The vast majority of LMICs lack sufficient mental health specialists (Kohrt et al., 2015), and cannot afford to allocate substantial proportions of health budgets to mental health interventions that require complex training and multiple sessions for disorder-specific protocols (Singla et al., 2017). An additional barrier for lengthy treatments in LMICs is that this can be a barrier for people to engage with them because they often have competing events that preclude attendance at lengthy interventions (e.g., the need to find employment, child care duties) and transport to centres where intervention is offered may be costly or dangerous.

These factors preclude many LMICs from implementing mainstream psychological interventions commonly recommended in high-income countries for common mental disorders. This has led to a shift towards brief scalable interventions that rely on training local non-specialists to deliver mental health programs. These tend to be more transdiagnostic in nature in order to minimise the need for local providers to conduct complex assessments and make differential diagnoses. This task-sharing approach has been embraced broadly in global mental health, and there is an increasing evidence-base for these interventions, with one meta-analysis of 27 trials reporting a pooled effect size of 0.49 (Singla et al., 2017).

It is important to note that scalable interventions can also be required in high-income countries that are hosting refugees. Many countries that have excellent health systems and have ample mental health specialists may have inadequate mental health providers who can speak the appropriate language of refugees or be culturally acceptable to refugees (Kiselev et al., 2020). Further, there can be challenges in terms of accessing care due to health insurance barriers, waiting times or costs. These factors underscore the need to not restrict consideration of scalable interventions for refugees to LMICs but also as a potential means of addressing mental health needs in health systems that are otherwise well-resourced.

This review now turns to consider some of the major types of scalable interventions that are supported by evidence. As noted earlier, the selection of interventions and studies reviewed are not the result of a systematic review (see Singla et al., 2017) but rather certain interventions are selected that exemplify the different types of interventions that aim for scale-up.

## Transdiagnostic interventions

In recent years the World Health Organization (WHO) has embarked on a series of scalable interventions, the first of which was Problem Management Plus (PM+; World Health Organization, 2016), which is a 5-session program that adopts a transdiagnostic approach to reducing common mental disorders such as anxiety and depression. It focuses on four basic strategies that include arousal reduction, problem-solving, behavioural activation, and accessing social support, and presumes eight days of training to lay providers (Dawson et al., 2015). This program was initially shown to be effective relative to enhanced usual care when delivered to people affected by adversity in both individual formats in Pakistan (Rahman et al., 2016) and Kenya (Bryant et al., 2017), as well as when delivered in small group formats in Pakistan and Nepal (Rahman et al., 2019; Jordans et al., 2021). These trials involve a combined total of nearly 2,500 people, highlighting the robustness of the efficacy of this intervention.

The extent to which PM+ was effective for refugees has only been more recently tested. Several pilot trials were initially conducted to determine whether individual PM+ (in the Netherlands, de Graaff et al., 2020; in Switzerland, Spaaij et al., 2022) and group PM+ (in Turkey, Acarturk et al., 2022b) administered to Syrian refugees was acceptable and safe. Another pilot trial (that was not fully controlled) with Venezuelan migrants and refugees found at posttreatment that a culturally adapted version of PM+ resulted in greater psychological well-being relative to wait-list controls (Perera et al., 2022). These trials, which were conducted in preparation for fully-powered controlled trials each reported high levels of acceptability, attendance at sessions, and lack of adverse reactions. One fully-powered trial of group PM+ has been reported with refugees in which Syrian refugees in a camp in Jordan were administered group PM+ or enhanced usual care (Bryant et al., 2022a). This trial found that refugees administered group PM+ reported less depression at 3-month follow-up relative to the usual care. It is worth noting that this study did not report reductions in anxiety or PTSD symptoms, which has been reported in other trials of PM+ with non-refugee populations (Rahman et al., 2016, 2019).

## Self-help interventions

In an attempt to achieve greater scalability, there have also been attempts to promote self-help programs that rely less on a health provider and place emphasis on the person assisting themselves by giving them materials that instruct them on how to use strategies to achieve better mental health. One example of this approach is the WHO's Self-Help Plus program, which was initially trialled by delivering it to groups of 20–30 people by a facilitator who assisted participants as they worked through an illustration-based self-help manual (Epping-Jordan et al., 2016). This program (Self-Help Plus; SH+) borrows from Acceptance and Commitment Therapy to promote psychological flexibility, and teaches strategies such as cognitive diffusion, mindful practices and values clarification exercises (Hayes et al., 2011). One large trial has been conducted with South Sudanese refugees in Uganda, which showed that SH+ achieved a small significant effect relative to usual care in reducing psychological distress (Tol et al., 2020). Although this trial indicated yielded only a modest effect size for SH+, in the context of its capacity to train many people simultaneously this program has potential as a scalable intervention in settings with few health resources. Other trials have used SH+ for secondary prevention by providing to people with subsyndromal distress, with the goal of preventing onset of mental disorders. This approach has been used to limit onset of mental disorders in refugees. One trial of 642 Syrian refugees in Turkey found that refugees randomised to receive SH+ were significantly less likely to have a mental disorder six months later (21.7%) relative to enhanced usual care (40.7%) (Acarturk et al., 2022a). A different finding was observed in a multinational European study of 459 refugees that found that whereas SH+ did reduce frequency of mental disorders at the post-intervention

assessment, this effect was not evident at the primary outcome of 6-month follow-up (Purgato et al., 2021).

### Scalable face-to-face interventions

A range of programs has been successfully trialled that involve more than 5–6 sessions that the brief programs described above comprise. There are several popular protocols involving 12–16 sessions that employ lay providers who are briefly trained in the protocols. One intervention that has been repeatedly tested is the Thinking Healthy protocol, implemented that have focused on depression, including maternal depression (Rahman, 2007). This program trains lay providers to deliver cognitive behavioural strategies, including active listening, problem-solving, identification of unhelpful thinking patterns and utilisation of social supports. This program was developed to be integrated into the work routine of health workers caring for perinatal women, and comprises 16 sessions. Controlled trials have consistently shown it is effective in reducing depression in mothers in Pakistan (Rahman et al., 2008; Maselko et al., 2015; Waqas and Rahman, 2021). Despite the apparent success of this program, it has yet to be tested in refugees.

Another repeatedly studied program adopts a transdiagnostic approach by focusing on common elements of interventions that can benefit people with a range of mental health problems. This intervention, titled *Common Elements Treatment Approach,* intended to address common problems such as anxiety, depression and posttraumatic stress, and can target additional problems depending on the local need (Murray et al., 2014). CETA, which comprises multiple modules that address each target problem, relies on training lay providers to follow decision trees to determine which modules would be appropriate for each person. This approach differs from some other scalable interventions, such as some of the WHO programs, in that it is focused on individuals, and can be adapted for children and adults. For example, if alcohol abuse is indicated then an appropriate module comprising appropriate strategies would be employed. This protocol often involves 10–12 sessions of treatment, and the dosage of each module can vary according to a person's need. CETA has been shown to be effective in a range of trials in LMICs in refugees in reducing a range of mental health problems, including depression, anxiety and posttraumatic stress in Ukraine (Bogdanov et al., 2021) and Thailand (Bolton et al., 2014).

### Child and adolescent interventions

In contrast to the development of scalable interventions for adults, there is less known about their effectiveness with children and adolescents in LMICs (Fazel, 2018). One meta-analysis concluded that there was no robust evidence of psychological interventions for children or adolescents in reducing anxiety or depression (Purgato et al., 2018). This conclusion is reinforced by an umbrella review of nine meta-analyses of psychosocial interventions for children or adolescents in LMICs, which concluded that although there was some evidence for treatments of PTSD, there was no evidence for interventions to reduce anxiety or depression (Barbui et al., 2020).

A recent attempt to address this issue is the WHO's adaptation of their PM+ protocol, titled Early Adolescent Skills for Emotions (EASE). This program aims to reduce internalising problems, such as anxiety and depression, in 10–14-year-old adolescents (Dawson et al., 2019). The program comprises seven small group sessions for young adolescents that focus on arousal reduction, behavioural activation, and problem management as these strategies have been

shown to be key for shaping reducing internalising problems in adolescents. The intervention also comprises three group sessions for caregivers that teach coping skills, positive parenting, and inform them of the strategies taught to the adolescents. One initial trial of the EASE program randomised young adolescent Syrian refugees in Jordan to either EASE or Enhanced Usual Care, and found that three months after the intervention the adolescents reported greater reductions in internalising problems relative to those in usual care (Bryant et al., 2022b). Although this study indicated support for the EASE program, further studies are needed to validate the robustness of this effect.

One of the formats commonly applied in terms of scalable interventions for youth is school-based programs. This approach is often undertaken because schools represent a context in which most young people in LMICs can be reached and promoting better mental health has the potential benefit of increasing school attendance and academic progression. There is evidence that school-based programs can result in greater access to interventions relative to standard healthcare programs (Barry et al., 2013). There are systematic reviews indicating that school-based programs that comprise mental health promotion components can benefit youth, including refugees, mental health (Barry et al., 2013; Tyrer and Fazel, 2014). One review that focused on trials that recruited school children and/or adolescents who had mental health problems found mixed findings, with approximately only half of the studies identified reporting positive findings (Fazel et al., 2014). Interestingly, the strongest effects were for PTSD, with most studies reporting positive effects for the interventions for those youth exposed to traumatic events. It should be noted, however, that there is much variability in the programs offered, and most efficacious programs are focused on processing trauma memories (Tyrer and Fazel, 2014). Although there is evidence that school-based programs can produce beneficial effects, there is also much evidence from large trials that school-based programs do not result in positive outcomes in Nepal (Jordans et al., 2010), Sri Lanka (Tol et al., 2012), Burundi (Tol et al., 2014) and the United Kingdom (Kuyken et al., 2022). It has been noted that non-significant effects are observed in some of the better-controlled and larger trials, which suggests that universal intervention programs may not be the optimal strategy for enhancing mental health in children and adolescents (Cuijpers, 2022). Extrapolating from this body of evidence, it is reasonable to conclude that there is insufficient evidence regarding the extent to which school-based scalable interventions are effective for young refugees. Further, it is worth noting that many children in LMICs do not have access to schools, so school-based programs do not offer a useful context in which programs can be offered in these settings.

### Digital interventions

Attempts to make evidence-based mental health interventions more scalable to people in LMICs have unsurprisingly turned to digital mental health platforms because of their capacity to reach large numbers with minimum ongoing costs and personnel. Although adherence to unguided digital mental health programs tend to be poor (Christensen et al., 2009), guided programs that have internet or telephone assistance have been shown to be as effective as face-to-face interventions (Cuijpers et al., 2019). One recent attempt to evaluate a guided self-help digital program in a LMIC was the WHO's Step-by-Step program, which is comparable to PM+ in that it involves an internet-delivered 5-session program that instructs the user in psychoeducation, behavioural activation,

stress management, positive self-talk, strengthening social supports, and relapse prevention (Carswell et al., 2018). The program is supported by 'e-helpers' who provide weekly telephone or message-based support for the users. In a controlled trial in Lebanon of 680 people with depressive symptoms, it was found that Step-by-Step resulted in reduced depression, functioning problems, posttraumatic stress and personally identified problems relative to enhanced usual care (Cuijpers et al., 2022). Early piloting suggests that the Step-by-Step program has the potential for scale-up with refugees, however, there are reported barriers in terms of the extent to which refugees will persist with the sessions and achieve an adequate dose of the intervention (Burchert et al., 2018).

## Are scalable interventions cost-effective?

One of the common assumptions of scalable interventions is that they are cost-effective. A number of cost-effective analyses have been conducted of these interventions in LMICs. There is some evidence that task-shifting interventions can be both effective in improving mental health and also cost-saving (e.g., in India, Buttorff et al., 2012). Other studies from Pakistan have reported that scalable interventions in LMICs are costlier than usual care but they achieve more effective mental health outcomes (Sikander et al., 2019; Hamdani et al., 2020). When evaluating the cost benefits of scalable interventions in LMICs, one needs to consider the costs of training local staff and implementing the intervention relative to the savings for the local health system. Reflecting the potential for cost-effectiveness of scalable interventions, one analysis of SH+ intended to prevent mental disorders in refugees in Turkey found that an outlay of $US2802 would result in a 97.5% chance of being cost-effective relative to enhanced usual care; this can be considered as highly cost-effective on global standards (Park et al., 2022). In summary, the available evidence suggests that these interventions can be cost-effective when one considers the gains achieved in health costs and other resource-demanding expenses associated with having mental disorders.

## Challenges for scalable interventions

### Non-responders to interventions

Despite the reported success of many task-sharing interventions for refugees in LMICs, it is important to recognise that many people in the reported trials do not respond positively to the intervention. It is usual practice in most trials to report an effect size of an intervention, and most studies positively appraise an intervention if it has been able to achieve a small or moderate effect. This pattern is reflected in meta-analytic studies of task-sharing interventions (Singla et al., 2017). These studies tend not to report the exact proportions of people who do not respond positively to the intervention. One can gain some insight, however, by calculating the proportion of participants who still have probable disorder at the follow-up assessments. For example, one trial of PM+ in Pakistan that yielded strong effects nonetheless reported that 26.9% of those who received PM+ still had probable depression at follow-up (Rahman et al., 2016). Another trial of PM+ provided to women who were survivors of gender-based violence in Kenya reported that at follow-up 21.1% of participants still had probable psychological disorder (Bryant et al., 2017). This is underscored by findings from studies with refugees, with one trial of PM+ in refugees

finding no reduction in anxiety relative to enhanced usual care (Bryant et al., 2022a).

There is a need for frameworks that can both address the persistent mental health needs of refugees in LMICs but at the same time be scalable in poorly resourced settings. One approach that has intuitive appeal involves stepped-care models (Patel et al., 2007). One version of stepped care can triage people to either brief, transdiagnostic interventions if they present with less severe psychological distress or to more intensive interventions if they experience more severe problems. Another variant of stepped care is to provide all people who are distressed with brief, transdiagnostic intervention, and if people have residual problems after the intervention they can be offered further more intensive interventions. Both approaches attempt to address the issue of meeting more severe or persistent problems whilst at the same time limiting the demands on local health resources. Stepped-care models have been implemented successfully in LMICs to address more severe mental health needs in LMICs (Araya et al., 2003; Patel et al., 2010). It should be noted that the effectiveness and cost-effectiveness of stepped-care frameworks relative to single mental health programs have not been formally evaluated in LMICs, and this remains a challenge for advancing scalable interventions in LMICs.

### Need for long-term follow-ups

One of the major limitations of our current knowledge of scalable interventions is that little is known of the longer-term effects of interventions. The vast majority of trials report only short-term follow-ups after the intervention, with most reporting outcomes between 3 and 6 months (Turrini et al., 2019). This is an issue in the context of scalable interventions in LMICs because most people in these settings experience many ongoing stressors, ranging from war, conflict, interpersonal violence, disasters and poverty. It is questionable if brief transdiagnostic interventions can achieve longer-term mental health gains in the context of significant ongoing stressors. Considering the ongoing stressors experienced by many refugees and the deleterious effects these can have on their mental health, it is important to determine if recommended treatments are beneficial in the long term. Initial evidence from a 12-month follow-up of PM+ with Syrian refugees found that initial benefits in reducing depression and personally identified problems observed at three months were not maintained at 12 months (Bryant et al., in press). There is a need to gain further understanding of the longer-term effects of scalable interventions because it is possible that booster sessions or other ongoing supports may be needed to sustain the benefits of initial provision of an intervention. Additionally, given that many refugees experience ongoing stressors and that these contribute to poor mental health (Miller and Rasmussen, 2017), interventions that aim to reduce these stressors through social and other contextual means may also play a key role in maintaining the longer-term effects of scalable interventions.

### Role of comparator interventions in research

Another area that requires careful consideration when evaluating the merits of scalable interventions is the comparator conditions that are used to measure the efficacy or effectiveness of the intervention. Determining the effect size of any intervention is dependent on how much an outcome is changed by the intervention *relative to* the comparison intervention. Using a benign comparison condition will yield a stronger effect for the intervention condition, whilst a comparator condition that contains elements

that are known to be helpful will diminish the relative effect of the intervention (Gold et al., 2017). Evaluations of scalable interventions have tended to use a variety of comparison conditions, including wait list (Weiss et al., 2015; Bogdanov et al., 2021) and treatment-as-usual comparison (Rahman et al., 2016; Bryant et al., 2017, 2022a) groups. This has resulted in varying tests of the efficacy of interventions. Moreover, many research designs have not allowed the specification of specific versus generic therapeutic effects of treatment. For example, comparing group PM+ with treatment, as usual, does not delineate between the effects of strategies taught in group PM+ and the nonspecific effects of attention from a counsellor and group involvement. Highlighting the importance of testing the specific effects of scalable interventions is a recent study conducted during the COVID-19 pandemic that found that SH+ did not improve mental health outcomes relative to an alternate intervention that was comparably structured (Riello et al., 2021).

### Implementation studies

The extent to which interventions are truly scalable relies on them being evaluated in implementation rather than simply efficacy trials. Most interventions that are described as scalable are initially tested in reasonably controlled environments and under strict scientific conditions. Most trials to date of scalable interventions have employed strict training of lay providers, often with competency checks and rigorous supervision throughout the trial. Others may enhance motivation of participants to comply with repeated assessments by offering cash or in-kind reimbursements for their time. Many trials are also led by centres of excellence that are not necessarily representative of normal health care in LMICs. Further, these trials are supported by substantial research grants that permit a level of resources to the delivery of the intervention that may not be present in real-world implementation. The next stage of research in determining the effectiveness and cost-effectiveness of scalable interventions is to monitor these interventions when they are implemented in local healthcare systems. Under these real-world conditions, it will be essential to evaluate how the gains in mental health achieved in efficacy trials translate to implementation research.

Relatedly, at present, we do not have accurate data on the extent to which evidence-based scalable interventions are being implemented in government health systems and non-government organisations. One recent review noted that few scalable interventions have been scaled up in the local health system, and noted that lack of awareness of mental health programs, poor consensus on how implementation may operate, lack of political will to initiate implementation, and the distribution of financial resources to mental health services all contribute to implementation not occurring in local health systems (Troup et al., 2021). The goal of developing scalable interventions is to provide programs that poorly resourced countries can implement for people with mental health needs, so it is critical to evaluate the extent people in these settings are trained in these programs, are implementing them in regular health services, and also evaluating their effectiveness. To this end, implementation science paradigms offer useful metrics to assess the extent to which scalable interventions are being adopted in policy and practice documents, as well as being implemented at a practical level (Powell et al., 2015; Charlson et al., 2019a, 2019b).

One of the potentially useful ways forward in promoting greater implementation of scalable interventions is to promote locally-based intervention research. Much of the evidence-based for scalable interventions have come from studies initiated and led by researchers based in high-income countries, who subsequently collaborate with local implementing partners. This pattern may be a hindrance for implementation because the driving force of the intervention in the specific setting has not been local health providers or academics. Promotion of local expertise in leading trials, and especially implementation trials, may facilitate greater implementation into local health services after trials are complete.

### The extent of need for mental health services

One of the challenges in implementing scalable interventions in LMICs is that in many countries there is inadequate knowledge of the extent and nature of mental health needs. This is an important omission for public health initiatives in planning and implementing mental health services because it is essential to know the frequency of people with mental health problems, the type and severity of mental health problems, the subgroups of people who are most in need, and the extent to which people will seek help that is offered. Although we have many studies that indicate that refugees and Internally Displaced Persons (IDPs) in LMICs experience high rates of mental health disorders (Blackmore et al., 2020), there are many gaps in many countries that cannot afford to map the specific mental health needs by conducting representative sampling of the population. Even in countries that have been able to achieve this, there is always the need for ongoing monitoring of needs because of contextual changes.

### Concluding comments

In summary, the last decade has seen a surge in research in scalable interventions in LMICs that train lay providers to deliver a wide range of programs aimed to improve mental health. Overall, the findings have been very promising in that they suggest that much can be achieved using local resources to deliver brief and potentially affordable interventions. Despite these gains, there is much work to do to have sufficient evidence to guide governments and agencies on how to optimally implement these interventions in settings with few resources. We need a better understanding of the longer-term effects and costs of programs, how to manage people who do not respond to interventions, and importantly, how mental health interventions interact with other social and health determinants that drive a person's mental well-being.

**Open peer review.** To view the open peer review materials for this article, please visit https://doi.org/10.1017/gmh.2022.59.

**Data availability statement.** Not applicable as no data is involved in this study.

**Author contributions.** R.A.B. wrote the manuscript.

**Financial support.** No funding to declare.

**Competing interest.** The author declares none.

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
