## [Reviewer Report]

20th June, 2022

Professor Gary Belkin

Editor

Global Mental Health

Dear Professor Belkin,

Please find attached a manuscript titled “Scalable Interventions for Refugees”. This review is an invited overview of the field of scalable interventions for global mental health. 

Sincerely,

Richard Bryant, PhD, DSc

---

## [Reviewer Report]

*Comments to Author*: This review provides a current overview on scalable mental health interventions in refugee populations, written by one of the leading experts in the field. This was an enjoyable read. The review is concise yet provides sufficient detail to understand where the field is currently at in terms of scalable mental health interventions. This will be a nice and valuable contribution to the field. However, I have a few suggestions and comments:

1. The sentence on page 4 starting with “Despite these high rates of common mental disorders, …” is linking high rates of mental health problems with the majority of refugees being hosted in LMICs. The connection between these two facts is not very clear here. I guess the author wants to introduce the notion of refugees being hosted in LMICs but I think some rephrasing would make that clearer.

2. Generally, I’m wondering whether the issue of the treatment gap should be addressed and discussed a little more. For example, whilst most assume that the main reason for the treatment gap is the unavailability of services, it has also been shown that other factors contribute to the treatment gap (e.g., low uptake due to lack of interest in treatment, especially in the context of cultural stigma). This suggests that making services more easily available, may not necessarily be sufficient to bridge the observed treatment gap.

3. At the top of page 5 it is not always clear what the % mean. Some clarification would be helpful.

4. On page 5, the end of the sentence in the middle of the page starting with “Meta-analyses indicate that there is moderate …” is not very clear. Does this refer to LMICs or high income countries? The sentence suggests both.

5. NET is mentioned on page 5 but not included in the evaluation of scalable interventions. Should NET developed for LMICs not be considered a scalable intervention and therefore be included when reviewing the efficacy of scalable interventions? If so, it would be helpful to clarify why not.

6. Generally, it would be helpful to indicate in the abstract and introduction, that this is not a systematic review and not all available interventions are included (i.e., there is a strong focus on WHO interventions). It would be good to indicate somewhere the criteria by which interventions were selected for inclusion (e.g., more recent interventions, or those evaluated through RCTs etc.).

7. On page 10, it may be helpful to provide a little more information on CETA, and how it differs from the WHO interventions (e.g., focus on individuals rather than groups, and application to both children and adults, inclusion of a parenting component etc.)

8. On page 12, I was wondering whether it would be possible to provide more information on the types of school interventions. I think IDRAAC in Lebanon have been running some programmes. IRC was running an evaluation of a programme and didn’t find much evidence of a positive effect. However, I don’t know whether these studies have been published yet. One of the important issue here to mention is that many children in LMICs do not have access to school.

9. Amongst the challenges from page 12 onwards, it would be good to mention the fact that it is often not known how big the actual demand for services is in these settings.

10. On page 15, a further suggestion could be (given on-going stressor in humanitarian settings) that mental health services need to be combined with other support aimed at alleviating on-going stressors.

11. I was also wondering whether it would be possible to provide some information on the uptake of the current scalable interventions in the field. In other words, how many of these interventions are being provided by organisations independent of on-going research studies?

12. Maybe at the end of the manuscript, it would be helpful to suggest future directions in research to overcome the highlighted gaps?

Thank you for the opportunity to review your manuscript. I hope the comments above

will be helpful.

---

## [Reviewer Report]

*Comments to Author*: It was a privilege to review this manuscript. The author shared important critiques on available interventions for refugee populations and recommendations to scale these interventions in LMICs. It is appreciated that the author acknowledged that interventions designed for use in LMICs by refugees are based on research completed in HICs. With limited intervention design research based in LMICs and its population, can you discuss how such research could facilitate or impede the scalability of mental health interventions for refugees?

Two separate abstracts with separate aims were provided. Therefore, the review below was guided by the aims provided in the introduction.

Sample: Understandably, much of the research presented were based on European refugees (e.g., Syrian) within European countries. It would be helpful for the reader for the author to define the refugee sample(s) that much of the cited intervention research were based on and provide a brief history rationale.

There was no mention of the recent intervention research for refugees in Latin America (e.g., Perera et al. 2022). With recent recognition by global organizations of refugees from Latin American LMICs (e.g., Northern Triangle, Venezuela), it is important to incorporate this limited but important research and discuss its potential for scalability in Latin America.

Prevalence: It is unclear why the prevalence of common mental disorders (CMDs) in LMICs is supported by research that determined the prevalence of CMDs for refugees in HICs (e.g., Henkelmann et al., 2020) or for refugees in both LMICs and HICs (e.g., Blackmore et al., 2020). Perhaps the author finds it important to provide the overall prevalence of CMDs for refugees in HICs and LMICs, as described in one of the abstracts. Therefore, to ease readability, it would be helpful to provide prevalence of CMDs for refugees in HICs and LMICs (e.g., Africa Europe, Latin America, Southeast Asia) separately.

The Need for Scalable Interventions: Page 3: “The need for effective mental health interventions for refugees is indicated by the strong evidence that refugees have higher rates of mental disorders than many other groups.” Do you mean “community sample?” Please clarify.

Available Mental Health Assistance: Please review the cited supporting evidence to ensure the manuscript offers accurate information. For example, Chisolm et al. (2016) estimated 7%-28% of people with depression, not CMDs, in LMICs and HICs received treatment; only 7%-21% of people with depression in LMICs only received treatment. The author is an expert in the subject matter, contributing and reviewing a considerable amount of literature, therefore it is very likely that the merging of evidence, such as this, was unintentional.

The Emergence of Scalable Interventions: Troup et al. (2021) provides an overview of policy/political barriers in LMICs the author may find important to note in this section.

Overall Organization: The manuscript was well written, and the author discussed the appropriate research within its respective sections. To improve readability, the author may consider how supporting evidence is presented within each section. Perhaps reorganize findings/literature by context (e.g., HIC vs. LMICs) and/or sample (refugees in HICs vs. refugees in LMICs).

---

## [Reviewer Report]

*Comments to Author*: Dear Prof. Marit Sijbrandij,

Thank you for the great opportunity of reviewing the manuscript titled “Scalable interventions for refugees”. The manuscript gives an overview of potentially scalable interventions for refugees mostly living in LMICs to treat mental illness and improve the mental health care of these populations. In general, the introduction is short and concise. It offers a good overview of the problem and research which will be addressed in the review. Further, it leads to the research questions addressed in the review. Still, in my opinion it would be helpful to clearly state the aim and scope of the review as well as the research questions addressed explicitly in the end of the introduction. Afterwards the author starts to introduce different categories of scalable mental health care interventions for refugees. The following categories are addressed: Transdiagnostic Interventions, Self-Help Interventions, Longer-Term Interventions, Child and Adolescents Interventions, and Digital Interventions. Hence, he addresses the currently most important categories in this field of research. For each category, he presents one or two examples and their current state of evidence. Thereby, the author introduces the category shortly. Afterwards, current evidence is presented citing different studies and meta-analyses. Although this form or presentation is very demonstrative for the readership, the methods applied remain unclear. For instance, it is not clear how and why the author selected the single interventions presented. Hence, the readership does not get a clear idea of the scope of the review and potential biases. A short paragraph on the methods applied indicating the scope and the limitations of the review might be helpful to orient the readership. Accordingly, it might be helpful to specify the type of review in the title to orient the readership early in the reading process.

One strength of the manuscript is the mainly current evidence cited and the number of meta-analyses and reviews mentioned in the manuscript. At the same time, some sentences are very long and complicated. Shortening some sentences might easily improve readability of the manuscript. Further, some paragraphs show a lack of citations. I would recommend to add some citations to strengthen the argumentation. For example, in the paragraph on digital interventions the author claim that unguided versions are not effective without citing evidence.

Summarized, the manuscript offers a great overview of the current state of research on scalable interventions for refugees. Adding some information on the aim and methods applied in the review process might improve the manuscript giving the readership a comprehensive picture of the aim and limitations of the review presented.

---

## [Reviewer Report]

*Comments to Author*: Many thanks for submitting this very informative narrative review to GMH. It was a pleasure to read. In addition to providing a concise overview of the evidence available on scalable interventions, you also outline the most pressing challenges for scalable interventions, which provides very relevant and practical suggestions on how to move the field forward.

In addition to the suggestions made by the three reviewers, please find below a few additional points.

1. Introduction, page 4: you cite the meta-analysis on prevalence of mental disorders of Blackmore and colleagues. You may also include a more recent meta-analysis published in GMH, that also differentiates between refugees in LMICs v.s in HICs (also relevant to the point regarding prevalence raised by reviewer 2): 

Patanè et al (2022). Prevalence of mental disorders in refugees and asylum seekers: a systematic review and meta-analysis. Global Mental Health, 1-14.

2. Under the category "longer-term interventions" you describe in-person delivered interventions for adults, with higher intensity than the self-help interventions. Please consider to rename this category (for example "Scalable face-to-face interventions for adults"). Note that the intervention developed by Atif Rahman is named Thinking Healthy (instead of Healthy Thinking).

3. In terms of the effects of school-based interventions, studieus have produces rather mixed results. Studies in Burundi, Indonesia, Nepal, and Sri Lanka (Jordans et al., 2010; Tol et al., 2014; Tol et al., 2012; Tol et al., 2008) found positive effects, though not on all outcomes and for all children, and sometimes with negative effects for vulnerable groups. In addition, they are not effective as universal prevention strategy (see Cuijpers, P. (2022). Universal prevention of depression at schools: dead end or challenging crossroad?. Evidence-based mental health, 25(3), 96-98.)

---

## [Reviewer Report]

20th October, 2022

Professor Marit Sijrandij

Handling Editor

Global Mental Health

Dear Marit,

Please find attached a revised manuscript titled “Scalable Interventions for Refugees”. I have responded to each of the reviewers’ comments. 

Sincerely,

Richard Bryant, PhD, DSc

---

## [Reviewer Report]

*Comments to Author*: All my comments have been addressed and the manuscript revised accordingly. I have no further requests.

---

## [Reviewer Report]

*Comments to Author*: General Comments: I thank the author for the improved review. It is an important review, incorporating previously published work (for example, Morina et al., 2017a; Morina et al., 2017b) to support his notions. This review will be helpful for early career researchers to identify limitations and considerations in global mental health research. To elevate this review, more of the author’s critical review is needed in the “Challenges for Scalable Interventions” as well as a section offering guidance for researchers in their upcoming research.

Below are my comments.

Page 5: Please clarify this sentence: “Representative studies of refugees reported estimated prevalence of 15% of bereaved refugees from various backgrounds in both a high-income (Bryant et al., 2019) as well as Syrian refugees in a LMIC (Bryant et al., 2021) settings.” Are the bereaved refugees from HICs or are the Syrian refugees from high-income backgrounds or both?

Page 6: Please clarify this sentence: “It is estimated that mental health assistance for people with depression around the world ranges from 7% to 28%, with treatment provision being only about one-third of cases in LMICs compared to more than half of cases in high-income countries (Chisholm et al., 2016).” There is a range for prevalence, yet it is unknown the exact proportion of the exact prevalence rate is getting treatment. Please be more specific.

Page 6: Please provide the total numbers for each context (i.e., LMIC, HIC) as it may be that HICs had less refugees compared to LMICs, which can explain the difference in estimates. “One review found that whereas 36.3% of respondents in high-income countries in the World Mental Health Survey who reported an anxiety disorder received help, only 13% of those in LMICs reported receiving assistance (Alonso et al., 2018).”

Page 6: This statement is vague. “Another meta-analysis of global studies found a significant gap in health service use, which ranged from 51% of those needing help actually receiving it in high-income countries to 20% in LMICs (Moitra et al., 2022). This meta-analysis also reported that whereas 23% of people received minimally adequate treatment, only 3% received this level of care in LMICs.” What type of “health service?” 23% and 3% received “minimally adequate treatment” in HIC and LMICs, respectively. What is this referring to? Mental health treatment or medical health treatment?

Page 6: The paragraph with the main sentence, “The lack of mental health services is not the only barrier to people accessing mental health care in LMICs” does not include a statement that is not pathologizing or patronizing to refugees’ decision to not receive mental health services. It is important to note, perhaps along with or after this statement [“Many refugees also have negative beliefs about receiving help for mental health problems, which limits their motivation for seeking help that may be available (Byrow et al. 2020).”] that refugees may have their own methods in managing distress and cite relevant work. It is necessary to acknowledge refugee’s inherent coping skills which can explain low rates of treatment-seeking.

---

## [Reviewer Report]

*Comments to Author*: Thank you for the opportunity to review the revised manuscript on Scalable Interventions for Refugees. I believe that the author has carefully considered the reviewers' comments. In the process, the manuscript has improved significantly from the first version. The scope and methodology used are now much clearer and additional recent literature has been included. The examples presented for each category of scalable interventions present the current state of research, providing important insights for the readership. At the same time, the author clearly identifies current limitations, knowledge gaps, and future challenges. This helps the readership to quickly gain an overview and derive relevant next steps in research on scalable interventions for refugees.

The only things I noticed while reading the manuscript were some small typos and inconsistent citation of the literature. This could be fixed as the publication progresses.

Once again, I appreciate the opportunity to review your manuscript.

---

## [Reviewer Report]

*Comments to Author*: The author did a great job in addressing all issues raised by the reviewers, and including them in the revised version of this manuscript. Reviewer 3 had a few additional comments, which the author may address in a minor revision.